# The Importance of the Microbiota in Shaping Women's Health—The Current State of Knowledge

Karolina Krupa-Kotara [1,*], Paulina Helisz [1], Weronika Gwioździk [1] and Mateusz Grajek [2]

1   Department of Epidemiology, Faculty of Health Sciences in Bytom, Medical University of Silesia in Katowice, 41-902 Bytom, Poland
2   Department of Public Health, Faculty of Health Sciences in Bytom, Medical University of Silesia in Katowice, 41-902 Bytom, Poland
*   Correspondence: kkrupa@sum.edu.pl

**Abstract:** According to current knowledge, a properly colonized human microbiota contributes to the proper functioning of the body. The composition of the natural flora changes depending on age, health, living conditions, and the use of antimicrobial agents: antibiotics, disinfectants, and some cosmetics. The human body is diversely populated with microorganisms and undergoes constant changes under the influence of various factors, and its proper composition is extremely important for the proper functioning of the body. Given the above, it was decided that we would review current scientific research that explains the cause–effect relationship between the composition of microorganisms populating the human body and health, focusing on women's health. As a result, an overview paper was prepared based on 109 scientific sources from 2009–2022. Special attention was paid to the most recent scientific studies of the last five years, which account for more than 75% of the cited sources.

**Keywords:** microbiome; women's health; microbiota

## 1. Introduction

The topic of intestinal microbiota is becoming of interest to an increasing number of researchers around the world. Each year, database search engines note an increase in published papers in this field. According to the current state of knowledge, a properly colonized human microbiota contributes to the proper functioning of the body. I consider the microbiota to be microorganisms (bacteria, fungi, protozoa, viruses) that permanently or transiently colonize selected areas of the human body. All microbiota form a structure in the human body called the microbiome [1]. In turn, the microbiota is defined as the totality of all microorganisms that can colonize the skin, respiratory system, genitourinary system, and—primarily—the digestive system. A proper balance between bacterial strains and the human body allows for proper homeostasis [1,2]. It is estimated that the mass of the microbiome, or the mass of the entire microflora of the human body, exceeds 2 kg. Colonization of the newborn with microorganisms from the mother (genital tract, oral cavity, skin), medical personnel, and the hospital environment begins immediately after birth. Depending on the birth type, the newborn's gastrointestinal tract is colonized by different microorganisms. In the case of children born via natural childbirth, microorganisms from the mother's vagina and digestive tract—during caesarean section, microorganisms from the mother's skin and medical personnel [3]. In shaping a woman's microbiome, it is also worth relying on the concept of intrauterine programming, which specifically influences the state of the fetal gut microbiota through regulation of the gut–brain–lung axis. Therefore, such a method of prenatal nutrition should be considered in the future for chemopreventive and immunomodulatory effects on the microbiome. It is worth extending diagnostics with prenatal testing and detailed medical history with both parents as a preventive measure. Supplementation during fetal development and natural childbirth, as well as the method

of feeding and the lack of diagnosis of metabolic diseases (e.g., obesity, hybrid diabetes, various endocrinopathies, MTHFR polymorphism) with a balanced dietary regimen during adolescence, predisposes the microbiome to function properly in adulthood [4].

There are differences in the quantitative and qualitative composition of bacteria, fungi, and archaea residing in different digestive tract sections. The number of bacteria residing in the intestines increases compared to the number of microorganisms inhabiting the oral cavity and stomach. In the jejunum, there are bacteria mainly of the genus Bacteroides, Lactobacillus, and Streptococcus—in the ileum, mainly Bacteroides, Clostridium, Enterococcus, Lactobacillus, and Enterobacteriaceae. In the large intestine, there is the most numerous set of microorganisms, totaling about 1.5 kg to 2 kg. Mainly, the bacteria are Firmicutes, Bacteroidetes, Proteobacteria, and Actinobacteria [5]. The task of microorganisms inhabiting the gastrointestinal tract is to limit the growth of pathogenic bacteria as well as to participate in the digestion of food and to participate in the synthesis of vitamins, the production of short-chain fatty acids—SCFAs—which are formed via the fermentation of exogenous complex carbohydrates and contribute to the regulation of energy balance. SCFAs are the most important source of energy for colonocytes and include butyric acid, which stimulates maturation and proper differentiation of colonocytes, reduces concentrations of pro-inflammatory cytokines, and has a beneficial effect on the continuity of the mucosal barrier [6]. The task of the intestinal microbiota is to coordinate and activate the immune system and its metabolic role, which plays an important role in maintaining the body's homeostasis. The intestinal microbiota is also involved in the biosynthesis of many vitamins necessary for the host body, such as vitamin K, B vitamins (B1, B2, B3, B6, B12), and folic acid, as well as in metabolizing undigested food residues, from which the body derives additional energy. The type, species, and strains of bacteria inhabiting the intestines depend on many factors, i.e., the type of delivery, type of natural or artificial infant feeding, age, place of residence, intestinal pH, diet, and medications taken [5,7–9]

The composition of the natural flora changes depending on age, health status, living conditions, and the use of antimicrobial agents: antibiotics, disinfectants, and some cosmetics. The human body is colonized with microorganisms in a differentiated manner, which makes it possible to distinguish the following areas:

- The area having permanent colonization—skin, mucous membranes of the upper and lower respiratory tract, upper and lower gastrointestinal tract (especially oral cavity and large intestine), vagina;
- The area having little transitional colonization—larynx, trachea, bronchi, lateral sinuses of the nose, the middle section of the gastrointestinal tract (esophagus, stomach, the upper part of the small intestine, urethra, cervix, conjunctiva;
- The non-colonized area—bronchioles, alveolus, tears, blood, cerebrospinal fluid, urine, tissues, and tissue fluids [10].

The relationship that occurs between the microorganisms that make up the natural flora and the organism can take the form of mutualism (symbiosis), a mutually beneficial coexistence. A classic example is the presence of *Escherichia coli*, which are part of the intestinal flora obtaining for themselves the substances needed for growth from digested food while taking part in the synthesis of certain vitamins, such as vitamin B12. Most often, however, the presence of natural flora brings neither benefit nor harm to microorganisms. This form of coexistence is referred to as commensalism [11].

This study aimed to investigate the hypothesis, relating to the impact of a healthy microbiome on the formation of women's health. To achieve this goal, the following research questions were posed:

Q1: Are there correlations between microbiota status and women's overall health?

Q2: Does a normal microbiome composition positively correlate with a reduced risk of developing diseases?

Q3: Does the presence of particular strains of bacteria condition the development of specific diseases?

## 2. Materials and Methods

### 2.1. Methodology Background

This study aimed at investigating the hypothesis relating to the impact of a healthy microbiome on shaping women's health. Current scientific research clearly emphasizes how crucial a role the microbiome plays in human health. Therefore, the scientific evidence was reviewed based on the available literature.

### 2.2. Review Procedure and Search Strategy

The following paper was edited based on good practices that are commonly used in works of this type. The authors of the paper began by defining the research field. To do this, they searched the PubMed database and found scientific publications consistent with the topic under consideration. The literature items were searched by the authors of the paper and a qualified library employee, using relevant keywords with Boolean operators and their combinations and configurations—gut microbiota, oral microbiota, skin microbiota, respiratory microbiota, reproductive tract microbiota, dysbiosis, and women's health—using the methodological tool in the form of the PubMed database.

### 2.3. Sources Selection

The literature search yielded many records, from which 2784 sources directly related to the topic of the paper were selected. Then, those with the highest scientific value were selected according to bibliometric impact factors. The final literature review was based on 91 sources, representing mainly scientific output from recent years (Figure 1).

The work's reliability, accuracy, and relevance were assessed using the GRADE (The Grading of Recommendations Assessment, Development, and Evaluation) system, one of the main goals of which is to eliminate confusion arising from the use of different evaluation methods. As a result, an overview paper was prepared based on 109 scientific sources from 2009–2022. Special attention was paid to the most recent scientific studies of the last five years, which account for more than 75% of the cited sources.

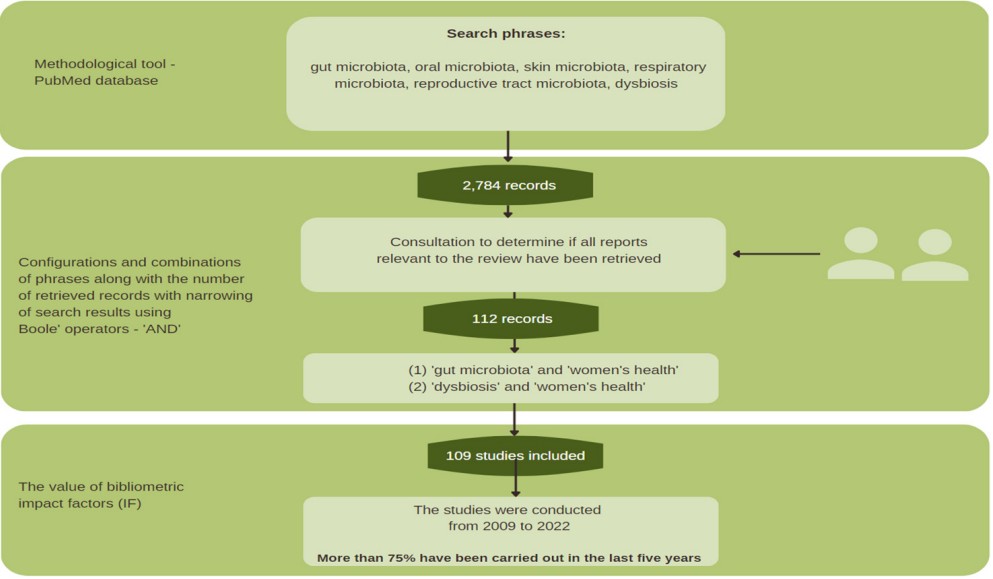

**Figure 1.** Methodological scheme.

## 3. Microbiota Distribution

### 3.1. Skin

The skin, being the largest human organ, plays an extremely important role in the immune system. It is the first line of defense both against changes in the external environment and against microbial attacks. Skin colonization depends on its moisture content and pH, and the number of bacterial cells (CFU—colony-forming unit) in 1 cm$^2$ varies from about 10$^4$ to 10$^5$. The natural flora of the skin consists primarily of gram-positive bacteria with a predominance of granulomas—*Staphylococcus epidermidis*, *Staphylococcus aureus*, and aerobic tentacles *Corynebacterium* spp. and *Propionibacterium acnes*—which are involved in the formation of juvenile acne [12–14].

In most areas of the skin, there are mainly gram-negative flora, and in the elderly, there are additionally fungi of the *Candida* family. The composition of the natural flora undergoes constant changes related to the secretion of glands and the coexistence of skin diseases and systemic conditions. The armpit and groin areas manifest increased sweat production. Areas such as the face and back are richly supplied with sebaceous glands. On the other hand, continuous exposure of the skin of the arms and feet to the temperature of the external environment contributes to their dryness. The skin microflora also depends on age and gender. The skin of the fetus in utero is sterile, after which its first colonization with bacteria occurs during natural childbirth or a caesarean section [13,15].

Bacteria of the *Propionibacteriace* subfamily predominate on the scalp, including around the nose, ears, and hair, with significantly fewer on the skin of the arms. On the other hand, the trunk and extremities—especially around the armpits, soles of the feet, navel, and popliteal fossa—are dominated by *Staphylococcus* and *Corynebacterium* bacteria and fungi of the *Malassezia* spp. genus. The chest, back, and occiput are populated in the greatest numbers by *Propionibacterium* spp. and various species of *Staphylococcus* spp. The surface of the forearm, which is a dry area, is dominated by mixed microorganisms represented mainly by the *Betaproteobacteriace* group and *Flavobacteriales*. Examination of the skin microbiome for the presence of viruses showed that like bacteria and fungi, viruses form both a permanent and transient composition of the skin microflora. Analysis of viral nucleic acid (DNA—deoxyribonucleic acid) sequences on the surface of the skin includes three predominant strains: *Papillomaviridae*, *Polyomaviridae*, and *Circoviridae*. This is in agreement with the fact that in most individuals, papillomaviruses are the most commonly found on the superficial layers of the skin. It is worth noting that eukaryotic viruses can contribute to the development of skin diseases, including cancer [14–17].

Physiological differences, including different hormones found in men and women, result in differences in the incidence of microorganisms on their skin. Adolescence appears to be a critical point in a person's life when the skin microbiota is remodeled. Due to increased hormone levels, which contribute to the production of extra sebum, there is a proliferation of lipophilic bacteria *Propionibacterium* spp. and *Corynebacterium* spp., and fungal bacteria *Malassezia* spp. The bacterial microbiota of the skin is also affected by environmental factors, occupation, clothing used, and the use of antibiotics. The use of cosmetics is also important, although the mechanism of their effect is not fully understood [16,17].

### 3.2. Oral Cavity

The oral cavity, due to its contact with air, food, and water environments, is a dynamic yet highly diverse and unique environment for microorganisms. Bacteria residing in the oral cavity are involved in the metabolism of nutritional products. The first colonization of this area begins immediately after birth, and the main source of bacteria is the mother. This area is first colonized by *Streptococcus salivarius*, *Streptococcus mitis*, and *Streptococcus oralis*, and in the next few months, gram-negative anaerobes *Fusobacterium nucleatum*, *Prevotella* melanogenic, and *Veillonella* spp. appear. At a young age, the human oral microbiota becomes very stable and is represented by bacteria of the genus *Streptococcus*, *Veillonella*, *Fusobacterium*, *Porphyromonas*, *Prevotella*, *Treponema*, *Neisseria*, *Haemophilus*, *Eubacteria*, *Lactobacterium*, *Capnocytophaga*, *Eikenella*, *Leptotrichia*, *Peptostreptococcus*, and also

*Propionibacterium*. It is assumed that there are more than 700 bacterial species in the human oral cavity, with only about 50% of the bacteria present known. These data are confirmed by the Human Oral Microbiome Database (HOMD), which not only presents the current nomenclature of the oral microbiota but also contains data based on phenotypic, phylogenetic, and clinical studies [16,18,19]. The high diversity of the oral microbiome is influenced by temperature, pH, oxidation reduction potential, salinity, and saliva, which, in addition to providing nutrients, removes metabolic products. In addition, it contains numerous enzymes, e.g., amylase, antimicrobial peptides, and even antibodies. The condition of the oral cavity depends on the state of hygiene of the host (tooth brushing, mouthwash). The saliva microbiome of people living in different geographic zones has 100 different types of bacteria, including about 40 as yet undescribed. The most common microbes are *Streptococcus*, *Prevotella*, *Veilonella*, *Neisseria*, *Heamophilus*, *Rothia*, *Porphyromonas*, *Fusobacterium*, *Scardovia*, *Parascardovia*, and *Alloscardovia*. It has been shown that the tongue, as a muscular shaft covered with mucous membranes, is also a site colonized by bacteria. In healthy people, pathogens from the genera: *Prevotella*, *Neisseria*, *Streptococcus*, *Heamophilus*, and *Fusobacterium*. Bacteria of the genera *Prevotella*, *Streptococcus*, *Veilonella*, *Actinomyces*, and *Leptotrichia* predominated in the diseased subjects. An important fact about the oral microbiome is the biofilm formed above and below the gums, which differs in composition from the bacterial microflora. The microorganisms present in the biofilm form an extremely organized and active structure, within which they work together to cause the breakdown of organic matter and the extraction of energy. In the supragingival plaque, there are mainly gram-positive bacteria, such as *Staphylococcus mutants* and *Lactobacillus* spp. On the other hand, gram-negative bacteria, such as *Actinobacillus* spp., *Campylobacter* spp., *Fusobacterium nucleatum*, and also *Porphyromonas gingivalis*, are present in the subgingival plaque. Bacteriophages have been found in the oral cavity, whose presence is associated with potential bacterial hosts. These mainly include *Aggregatibacter actinomycetemcomitans* phages, and their number is positively correlated with periodontal atrophy. It has been proven that oral bacteriophages can exist both as commensals and as pathogens. The ecosystem that is the oral microbiome is an excellent place for certain viruses to thrive: herpesviruses, including HSV (herpes simplex virus) and EBV (Epstein–Barr virus). Studies of the oral microbiome in healthy humans have proven the presence of such fungi as *Candida* spp., *Aspergillus* spp., *Cryptococcus* spp., *Fusarium* spp., and *Alternaria* spp. [10,16].

### 3.3. The Gastrointestinal Tract

More than a century ago, Russian Nobel laureate Ilja Iljicz Miecznikow hypothesized that the health properties of kefir are related to the presence of live bacteria in it which cause colonization of the intestine. The human intestinal microbiota develops early in fetal life. Moreover, the fact that the placenta is not sterile is emphasized. The creation of the intestinal microbiota is a dynamic process directly dependent on genetic factors, the microbiota of the mother, the type of birth, environmental conditions, as well as the diet used by the mother during pregnancy and, later, the host itself. In the first days of a baby's life, the large intestine is colonized by strains of bacteria such as *Escherichia coli* and *Enterococcus faecalis*, followed by *Bacteroides*, *Bifidobacterium*, and *Clostridium*. Literature data indicate that the human intestinal microbiota is not formed until about 2 years of age and that it undergoes continuous modification over the next 3–5 years. The type of birth is important in the creation of the intestinal microbiota. At the moment of rupture of the chorioallantois membrane, the child comes into direct contact with the microorganisms of the mother's vagina, thus inheriting the original microbiota of the mother and her ancestors. The microbiota of the gastrointestinal tract, due to the functions it performs and the specificity of its structure, is an extraordinary place for the development of microorganisms. It provides living and growing conditions for both commensal microorganisms and those supplied with food. The stomach, as one of the sections of the gastrointestinal tract, is a transitional reservoir of food, where wetting, dissolution, and also mixing of food content

with gastric juice takes place. The gastric juice itself is a mixture of proteolytic enzymes and hydrochloric acid and facilitates the absorption of nutrients [11,20–22].

Due to the presence of an acidic environment, the stomach is considered an essentially sterile and unfriendly place for microbial growth. However, with the discovery of *Helicobacter pylori*, which colonizes the gastric niche, attention has been drawn to the fact that the stomach is populated by a diverse micro-community: *Firmicutes*, *Proteobacteria*, *Actinobacteria*, *Bacteroides*, *Fusobacteria*, *Lactobacillus*, *Streptococcus*, *Veilonella*, and *Escherichia coli*. Numerous scientific studies confirm that as stomach acidity decreases, the risk of developing various diseases, including cancer, increases. It is currently estimated that the microbial environment of the stomach contains 101–103 CFU of bacteria, making it, along with the esophagus and duodenum, the least colonized section of the gastrointestinal tract. The gut microbiota ensures not only the continuity of the intestinal epithelium but also the homeostasis of the immune system while protecting the macroorganism from the adverse effects of pathogenic bacteria, including *Salmonella* spp., *Shigella* spp., *Staphylococcus aureus*, *Campylobacter jejuni*, *Yersinia enterocolitica*, and *Listeria monocytogenes*. It should be noted that the microbiota of the stomach is not fully understood—the study of the microbial inhabitants of this part of the gastrointestinal tract is extremely difficult because they are subject to a large selection error, which is due to interfering factors, as well as the inability to detect possible viruses, fungi, etc. [23–25].

The intestinal microbiome is formed by viruses that affect host homeostasis and condition intestinal immunity. Among them are viruses that infect host cells and bacteriophages that attack bacteria. The most widespread are single- and double-stranded viruses of the orders *Caudovirales*, *Podoviridae*, *Siphoviridae*, and *Myoviridae*. The intestinal microflora plays an important role in food digestion and energy absorption and are involved in vitamin production. Microbes colonizing the gut cause the breakdown of complex carbohydrates, which are the source of certain nutrients. By breaking down fiber and intestinal mucin, they provide a source of simple sugars and short-chain fatty acids. The genome of bacteria is much richer than that of the host (up to 100 times) so microorganisms provide humans with many enzymes and metabolic pathways. Fermentation carried out by the intestinal microbiota provides up to 10% of energy from food, participates in the regulation of body weight, and regulates the amount of body fat present in the body. The gastrointestinal microbiota contributes to strengthening the integrity of the intestinal epithelium, stimulating processes related to the production and secretion of secretory antibodies and cationic peptides with antibacterial activity. In addition, it is involved in the stimulation of mucin, which is a component of the phenomenon called immune ignorance, responsible for the lack of direct contact between bacteria and immune cells. The composition of the intestinal microflora also affects immune hemostasis by regulating the size of the lymphocyte population and also the ratio of Th1 to Th2 lymphocytes. Bacterial commensals directly protect the host system against pathogenic microorganisms, including *Escherichia coli*, *Salmonella* or *Shigella*, and *Clostridium difficile*, mainly by altering the qualitative and quantitative nutrients available in the gut. The intestinal microflora also plays an important role in the synthesis of vitamin K and B vitamins (including B12, B1, and B6), the circulation of bile acids, and the transformation of mutagenic carcinogens (heterocyclic amines and N-nitroso-compounds), the production of which increases in the intestines in the presence of a diet rich in red meat. In addition, gut microbes are involved in the synthesis of amino acids, including lysine and threonine [26–28].

Changes in the composition of the intestinal microbiota are referred to as dysbiosis, which in turn promotes the development of many conditions (irritable bowel syndrome, neuropsychiatric disorders, food allergies, etc.) and contributes to the expansion of inflammation in the body. The microbial environment of the large intestine varies from person to person, and the causes of disorders within this ecosystem include such factors as obesity, the use of antibiotics without medical indications, and a high-fat, processed diet. It is worth mentioning that increased inflammation in the human body can contribute to the development of conditions including metabolic, cardiovascular, and cancerous diseases [29,30].

Bacteria are involved in the production of biotin and folic acid and in the absorption of magnesium, calcium, and iron ions. In addition, they enable more efficient energy absorption by breaking down polysaccharides, which are unpalatable to humans in their primary form. Intestinal bacteria participate in the production of short-chain fatty acids, which are a source of energy for intestinal epithelial cells and thus have a beneficial effect on maintaining the continuity of the mucosal barrier and have anti-inflammatory effects by reducing the concentration of pro-inflammatory cytokines. In addition, they induce the synthesis of the aforementioned mucins that protect the epithelium from toxins and pathogenic bacteria, thereby stimulating the immune system to act. In recent years, an increase in the percentage of *Bacteroidetes* (e.g., *Bacteroides vulgatus*) and *Proteobacteria* (e.g., *Escherichia coli*) has been observed in the gastrointestinal tract of Crohn's disease patients. The cell membranes of these bacteria contain lipopolysaccharide, which strongly stimulates the immune system. In addition, a reduction in the percentage of *Firmicutes* bacteria and thus in the amount of butyric acid they produce was found. Imbalances in the microbiota may also affect the pathogenesis and course of diverticular disease and the incidence of obesity. People with excessive body weight have been found to have an increase in the percentage of *Firmicutes*-type bacteria relative to *Bacteroidetes*-type. The result of such a change in the intestinal microbiota is greater availability of energy extracted from food, as *Firmicutes* bacteria metabolize nutrients to a large extent, which predisposes to obesity. The microbiome has been shown to provide the human body with an additional 80–200 kcal per day [31,32].

An example of the correlation between the state of the microbiome and women's health is the effect of the microbiome on the incidence of cancer, primarily colon cancer. Bacteria that produce butyric acid or are mediated by it inhibit the growth of cancer cells and induce their apoptosis. On the other hand, bacteria such as *Escherichia coli*, *Clostridium perfringens*, *Bacteroides fragilis*, *Bacteroides vulgatus*, and *Enterococcus faecalis*—via the enzymes β-glucuronidase and β-glucosidase—re involved in the synthesis of toxic and carcinogenic compounds. The composition of the microbiome is also important in the development of an allergic reaction in people with a genetic predisposition. The development of allergy is associated with the dominance of the Th2-dependent response. Bacteria of the species *Bacteroides fragilis* have been shown to have a protective effect, not only contributing to the predominance of the Th1-dependent response but also via regulatory lymphocytes, inducing an anti-inflammatory response, thus limiting the development of diseases caused by an excessive Th2 lymphocyte response in the mucous membranes of the gastrointestinal tract and respiratory system. Emerging reports suggest that the onset of allergic symptoms in children is associated with a decrease in *Lactobacillus* and *Bifidobacterium*. The link between the microbiome and autism spectrum disorders is also being investigated. Attention has been paid to changes in the gut microbiota in children with autism. They were found to have a 10-fold increase in the number of *Clostridium difficile* bacteria relative to healthy children. It is likely that the effects of neurotoxins produced by these bacteria may contribute to the manifestation of some of the symptoms [33–35].

### 3.4. Respiratory Tract

The mucous membranes of the upper respiratory tract are in constant contact with the external environment through the process of breathing. Through the nasal cavity, pathogens are introduced into the airways with each breath. The microbiome of the upper respiratory tract is highly differentiated due to constant contact with the external environment, resulting in each person having a corresponding microbiome. In contrast, the lower respiratory tract, depending on its region and especially the lungs, shows a different microbiota composition [2,34]. In healthy individuals, the upper respiratory tract is populated mainly by *Propionibacterium* spp., *Corynebacterium* spp., and *Staphylococcus* spp. The keratinized squamous epithelium of the nostrils contains sebaceous glands that produce substances that promote the growth of lithophilic bacteria, such as *Propionibacterium* spp. Microorganisms of this type are capable of hydrolyzing fats, releasing free fatty acids, which lowers the pH promoting the growth of coagulase-negative staphylococci. In addition, humid conditions and the presence of oxygen in this area affect the growth of *Staphylococcus aureus* in this area [35,36].

The lower respiratory tract, trachea, and lungs are significantly different in structure and function from the upper respiratory tract. They are lined with ciliary epithelium and numerous secretory cells that release mucin, surfactant compounds, proteases, and immunomodulatory proteins, among others, which form an immune barrier. The immunity of this region of the airways is also determined by macrophages, T lymphocytes, and dendritic cells, including Largenhans cells. Hence, it was assumed that these defensive elements of the respiratory system were sufficient to maintain sterility in it [37].

### 3.5. Genitourinary System Respiratory Tract

The genitourinary system includes the organs of the genital tract and urinary tract. The microorganisms inhabiting this area of the human body exhibit great diversity, forming a dynamic and unique ecosystem. The normal vaginal biocenosis of mature women includes about 100 types of aerobic and anaerobic bacteria. The vaginal microbiota balances itself by undergoing quantitative and qualitative changes, and the ratio of aerobic to anaerobic bacteria is 2:5. The vaginal ecosystem includes bacteria from the genera *Corynebacterium*, *Streptococcus*, *Escherichia coli*, *Staphylococcus*, *Mobiluncus*, *Prevotella*, *Peptostreptococcus*, *Bacterioides*, *Porphyromonas*, *Eubacterium*, *Gardnerella*, *Bifidobacterium*, *Klebsiella*, *Fusobacterium*, *Sarcina*, *Listeria*, *Mycoplasma*, and *Ureaplasma*. In the postmenopausal period, atrophic changes called atrophy occur in the lower genitourinary tract due to estrogen deficiency. The composition of the vaginal microbiota can change under the influence of antibiotic use or agents with endocrine or immune activity and during pregnancy. The microorganisms that colonize the genital tract protect it from vaginal inflammation, while the microbial balance of the vagina depends on the influence of bacteria and their metabolic products, estrogen levels, and the pH value in the vagina. Disturbing it leads to irritation and makes the body more susceptible to vaginal infections. The vaginal microenvironment in healthy women depends primarily on age, hormonal changes in the body, hygiene habits, sexual activity, and even eating habits. The microorganisms that physiologically colonize the vagina protect it from pathogens and activate the immune response by producing various antimicrobial compounds. The microbiome of this area is dominated by lactic acid bacilli, which are more abundant in African American women than in Caucasian women. It is assumed that *Lactobacillus* spp. may provide a protective environment against pathogens that cause bacterial vaginal infections. In addition, a normal component of the vaginal microbiota is associated with increased defensins (vaginal antimicrobial peptides—AMPs), which show protective activity against bacterial vaginitis [38–40].

Group B *streptococcus* (GBS—Streptococcus agalactiae, group B *streptococcus*) is a gram-positive bacterium that is the most common cause of invasive infections among newborns. About 10–30% of pregnant women are carriers of GBS. The infection is transmitted to the newborn from a mother's vagina that is colonized by streptococcus. Children diagnosed with GBS infection have a multiplied risk of developing septicemia of streptococcal etiology. Statistically, it is estimated that one in five women is a carrier of GBS. Streptococcal colonization can be transient, intermittent, or permanent, which is why a microbiological test from the genital tract and rectum is performed at 35–37 weeks of pregnancy. This is the primary way to determine if there is a risk of GBS infecting the baby. During delivery, it is determined whether there are additional risk factors, and based on the totality of the data, a decision is made to administer an effective dose of antibiotics intravenously at least 4 h before delivery (intrapartum chemoprophylaxis is also possible). Children of GBS-positive mothers should be subjected to special monitoring, and if there are alarming symptoms, their diagnosis should be expanded to include microbiological examination of blood and cerebrospinal fluid, lung X-ray, and empirical treatment against *Escherichia coli* and GBS. Streptococcus agalactiae can also pose a risk to the mother by causing fetal membrane infection, endometritis, septicemia, and meningitis [38,39,41].

The microbial environment of the vagina is species-diverse, and the presence of certain microorganisms can cause inflammation in the genital tract. Bacterial structures such as the lipopolysaccharide of gram-negative bacilli, the peptidoglycan of gram-positive bacteria, and others are involved in the induction of pro-inflammatory cytokines and the formation of both cellular and humoral responses of the body. In the prevention of cervical cancer, it is important to detect precancerous conditions, taking into account the examination for pathogens that affect the progression of existing lesions. The etiological factor of cervical cancer is chronic infection with oncogenic types of human papillomavirus (HPV). The progression of dependent dysplastic lesions in the cervical epithelium depends on the coexistence of inflammation of the reproductive tract and the state of the immune system. Reduced numbers of lactic acid bacilli and the presence of G. vaginalis may contribute to the prolonged state of HPV infection due to the negative function of the strain, which refers to the breakdown of vaginal mucus by destroying the glycoproteins in it. Atrophic changes in the genitals and lower urinary tract, as well as the associated complaints, are mainly derivatives of postmenopausal changes, including low concentrations of ovarian-derived sex hormones—primarily estrogen—in addition to vasomotor symptoms. From the point of view of current scientific research, the greatest important aspect of the vaginal microbiota is attributed precisely to estrogen. This hormone in the vaginal epithelium contributes to the accumulation of glycogen, which in turn is the most important substrate for lactic acid. However, it should be noted that the enzyme $\alpha$-amylase has its share in the formation of lactic acid bacilli colonies, which—by dissimilating glycogen—contribute to the formation of simple sugars (mainly maltose, maltotriose). The entire genital tract and the lower urinary tract undergo menopausal atrophy. In the ovaries, folliculogenesis ceases, and their volume and mass decrease significantly. The surface of the ovaries becomes wrinkled and dull, and the weight of the organs drops below 10 g. The cortex becomes thinner and does not contain ovarian follicles, although they may be visible in transvaginal ultrasound (TVU—transvaginal ultrasonography) up to 5 years after menopause. The ovaries begin to be dominated by the medullary layer, the main hollow part with sclerotic blood vessels. Sometimes there is hypertrophy of the lining, which is not a pathological condition. After menopause, the hormonal function of the ovaries is significantly reduced—estrogen synthesis is minimal, and the ovaries become the main source of testosterone in the body at this time, producing about 50% of the total pool of this hormone, which explains the decline in the number of lactic acid bacilli during this period. In the fallopian tubes, there is a flattening of the lining epithelium and atrophy of the cilia. Secretion and peristalsis of the fallopian tubes decreases, and their lumina relatively often become overgrown. A typical change in the uterus is a significant shortening of the vaginal part of the cervix with narrowing and even overgrowth of the cervical canal [40,42,43].

The results of some studies suggest that an altered microbiota of the fallopian tubes and endometrium and, in particular, a reduced percentage of *Lactobacillus* bacteria may reduce the rate of successful in vitro fertilization procedures. A microbiota dominated by *Lactobacillus* bacteria appears to correlate with a higher likelihood of reproductive success, as evidenced by the fact that patients undergoing in vitro fertilization procedures had an endometrial microbiota with a ratio of *Lactobacillus* bacteria. Implantation problems in these patients worsened, especially if the number of *Gardnerella* and *Streptococcus* bacteria increased [38,39].

The vaginal environment is a very complex ecological system that changes with a woman's age. Its quantitative and qualitative composition depends on many individual and external factors. Many of these are subject to modification, but some are described as constant. The overall health and immune status of the body play an important role in maintaining vaginal health.

The composition of the vaginal microbiota will also vary depending on the physiological period and hormonal fluctuations. During sexual maturation, increasing estrogen levels correlate with an increase in *Lactobacillus* counts and a decrease in vaginal pH [44,45]. Interestingly, during menstruation, some adult women show a decrease in lactobacillus abundance and an overall increase in the diversity of the microbiota, which, as was pointed out earlier regarding the vaginal ecosystem, is an indicator of an increased risk of dysbiosis [46,47]. Taking contraceptives can disrupt these natural changes associated with the menstrual cycle. During pregnancy, on the other hand, there is an increase in the abundance of *Lactobacilli*; moreover, the vaginal microbiota of pregnant women appears to be more stable than that of non-pregnant women [48]. Quite surprising changes concern postmenopausal women. Namely, this period is characterized by a decrease in the number of beneficial bacterial strains and an increase in the abundance of *E. coli* within the vaginal ecosystem, thereby increasing the risk of developing dysbiosis. The decrease in the abundance of *Lactobacilli* in the postmenopausal period may be related to the physiological decrease in estrogen release [49]. Restoration of vaginal microbiota has been observed in women taking hormone replacement therapy [50].

### 3.6. Effects of Hormones on a Woman's Microbiome

The distribution of the gut microbiota varies according to a woman's life stage (childhood, adolescence, pregnancy, menopause, old age). The gut microbiota is known to contribute to the development of gastrointestinal diseases, such as irritable bowel syndrome, obesity, inflammatory bowel disease, and colon cancer, but the exact etiology remains elusive. Recently, gender differences in gastrointestinal diseases and their relationship to gut microbiota have been suggested. In addition, estrogen and androgen metabolism are linked to the gut microbiome. Since the gut microbiome is involved in the excretion and circulation of sex hormones, the concept of the "microgenome" has been proposed, indicating the role of sex hormones in the gut microbiota. However, further research is needed for this concept to be widely accepted.

The endocrine function of the reproductive system involves several hormones controlled by complex feedback mechanisms. The ovaries, adrenal glands, and adipose tissue produce estrogen. Estrogens produced in the body or taken in as food can be metabolized by gut microbes [51]. The resulting metabolites again affect the host. Sex hormones directly modulate bacterial metabolism through steroid receptors, including estrogen receptor beta. Meanwhile, the gut microbiome with β-glucuronidase activity deconjugates conjugated circulating estrogens excreted in the bile most often secreted by *Escherichia coli*, *Peptostreptococcus*, *Bacteroides*, and *Clostridia* [52]. Deconjugation enables the process of reabsorption of estrogens into the system. The deconjugated estrogens circulate and affect many organs, not only reproductive but also the skeletal, cardiovascular and central nervous systems, through estrogen receptors. Typically, estrogens bind to nuclear receptors, causing conformational changes [53].

Progesterone has many important functions in a woman's body, including being essential for becoming pregnant and maintaining a pregnancy, normalizing blood flow, regulating glucose levels, and regulating zinc and copper levels in the body. Progesterone is also strongly associated with intestinal function, as it affects intestinal sensitivity and motility at the level of prostaglandins [54]. Its adequate levels also affect metabolism and the growth of probiotic microorganisms in the gut [55,56].

The prevalence of unfavorable bacteria can disrupt androgen metabolism and underestimation of levels, such as testosterone levels, can occur, leading to varicose veins, bladder weakness, hair loss, muscle weakness, low libido, or osteoporosis. Some species of bacteria (including *Clostridium scindens*) can synthesize glucocorticosteroids (e.g., cortisol) into certain androgens (e.g., androstendione, testosterone) [57].

Pregnancy is a condition in which major hormonal and j microbiota changes are observed. There is an increase in estrogen, prolactin, and progesterone, which affects changes in the intestines of women. As a result of the increase in progesterone and prolactin, the number of *Proteobacteria phyla* and *Actinobacteria* increases, while the number of *Faecalibacterium* and other bacteria that produce short-chain fatty acids decreases [58,59].

During pregnancy, changes similar to metabolic syndrome occur, mainly in the third trimester. This is when weight gain, insulin resistance, and minor inflammation occur. Therefore, for the fetus to develop properly and for the birth to go well, there is an increase in the number of microorganisms that have strong anti-inflammatory properties [60,61].

The gut microbiome has been proven to interact with thyroid hormones and vice versa. It is in the gut, among other things, that the conversion of thyroxine (T4) to triiodothyronine (T3) occurs. In total, 20% of this conversion occurs with the help of intestinal bacteria. Another 70% occurs with the help of the liver, which is also sensitive to changes in the composition of the microbiome. Therefore, dysbiosis is a straightforward path to insufficient levels of thyroid hormones and hypothyroidism [62,63].

An abnormal microbiota leads to a softened gut. When this occurs, undigested proteins can enter a woman's body, and this leads to an immune system response that can result in various types of autoimmune diseases. For example, gliadin, a protein that makes up gluten, is molecularly similar to proteins that make up thyroid tissue, so with the involvement of certain genes, the immune system can also start attacking one's thyroid, which leads to Hashimoto's disease [64].

The prevalence of abnormal microbiota also leads to constant inflammation, and this impacts the adrenal glands, which secrete cortisol at all times, which simultaneously reduces active thyroid hormones [65,66].

Hypothyroidism itself slows down the entire metabolism, including gastrointestinal peristalsis. Constipation develops, and other digestive problems arise, especially those related to gastrointestinal motility, which itself is a substrate for microbiota disorders such as *Candida albicans* overgrowth or SIBO (small-intestinal bacterial overgrowth) [67,68].

In terms of the female population's health, attention is being paid to the growing problem of polycystic ovary syndrome (PCOS). Long-term health consequences include an increased risk of miscarriage and pregnancy complications. PCOS is often accompanied by hyperandrogenism, which is associated with metabolic dysregulation. Thyroid hormones and luteinizing hormone (LH) are responsible for the increased amount of androgens in a woman's body. Women suffering from PCOS have elevated levels of LH, which in turn contributes to the production of excessive androgenic hormones. Recent scientific reports suggest that intestinal dysbiosis may be responsible for the development of PCOS, including via testosterone, which may contribute to changes in the lower gastrointestinal ecosystem [69–71].

For the moment, scientific studies highlight changes in the abundance of bacteria residing in the gut of PCOS patients, mainly from *Bacteroidetes* and *Fermicutes* species. It is through the aforementioned bacterial species that adverse metabolic and immunological changes may occur, which are explained by the altered production of short-chain fatty acids. The composition of the intestinal microbiota in women with established PCOS is compared to that of obese women, but *Escherichia* and *Shigella* bacteria predominate in the group of women in question (PCOS) [70]. Metabolites of the gut microbiota, as well as the microbiota itself, can in turn lead to, among other things, activation of inflammatory pathways or proliferation of pancreatic β-cells, and this leads to the development of, for example, insulin resistance. At this point, a cause-and-effect relationship between dysbiosis and the occurrence of endocrine disorders can be identified [71]. It seems noteworthy that the gut microbiota of the endocrine system has a multidirectional effect. Bacteria residing in the gut can both produce hormones, such as those responsible for general happiness (serotonin and dopamine), as well as perform a regulatory function (glucocorticoids androgens) [72].

Estradiol, which regulates the female monthly cycle, significantly affects the composition of the vaginal microbiota. The female reproductive tract is mainly populated by bacteria of the genus *Lactobacillus*, whose number changes significantly depending on the day of the menstrual cycle. The lowest level of the hormone is observed during menstruation, at which time the abundance of *Lactobacillus* is markedly reduced, while the balance of microorganisms is restored in the late follicular and luteal phases [73]. Studies highlight the role of dysbiosis in the course of obesity. Estrogen plays one of the most important roles in terms of metabolic processes, this explains the fact that women in the menopausal phase have an increased risk of cardiovascular disease and also obesity. This is because estrogen, along with leptin and the gut microbiota, is deeply involved in the body's energy balance. The 2019 results on mice from the study by Acharya et al. show that both estradiol and leptin can contribute to the modulation of the gut microbiota in women. In addition, the researchers highlight the role of estradiol as a protective factor against obesity induced by a high-fat diet [74].

### 3.7. Association between Obesity, Microbiota Dysbiosis, and Neurodegenerative Pathogenesis

As previously described, intestinal dysbiosis is a major causative factor in various gastrointestinal disorders [22,23], which can result in increased levels of lipopolysaccharides, pro-inflammatory cytokines, T cells, and monocytes, causing increased intestinal permeability through the microbiota–gut–brain axis [75,76]. This results in the accumulation of misfolded proteins, damage to axons, and demyelination of neurons, which is an important aspect in the pathomechanism of neurodegenerative disorders: Parkinson's disease, Alzheimer's disease, multiple sclerosis, and amyotrophic lateral sclerosis [77,78]. Many studies have shown that taking probiotics can help maintain the integrity of the gut, thereby alleviating the above inflammation and preventing the induction of neurodegeneration [75–80].

Researchers highlight the potential role of the gut microbiota in the pathogenesis of Alzheimer's disease (AD). AD patients show a decrease in the abundance and diversity of *Firmicutes* and *Bifidobacteria* and increased numbers of *Bacteroidetes*, *Escherichia*, and *Shigella* [81]. Some potentially pathogenic microorganisms, e.g., *Escherichia coli*, produce amyloid, which affects bacterial adhesion to the intestinal wall. As a result of the progressive permeability of the intestinal barrier and the blood–brain barrier with age, microbes and amyloid can enter the CNS, resulting in accumulation in brain tissue. As a result of the dysbiosis present in people with AD, there may be the production of cytokines and bacterial metabolites that can enter the bloodstream and reach the brain, causing inflammation. In addition, people with AD have been observed to have deficiencies in neurotransmitters, such as norepinephrine and serotonin, which play a role in regulating cognitive functions [82].

Parkinson's disease (PD) is the second-most common neurodegenerative disease, after AD. People with PD have a reduced abundance of bacteria from the *Prevotellaceae family*, as well as *Blautia* and *Roseburia*, which act as anti-inflammatory agents and are responsible for maintaining the tightness of the intestinal barrier by, among other things, producing antimicrobial substances. On the other hand, there is an increased abundance of *Enterobacteriaceae bacteria*, which can be responsible for inflammation among other things. An increase in *Clostridium cocoides* bacteria is seen in the early stages of the disease, while *Lactobacillus gasseri* is seen in those with advanced disease [83]. The total bacterial count in PD is lower than in healthy individuals. People with PD also show deterioration in gastrointestinal motility due to bacterial overgrowth in the small intestine [82,84]. As a result of the reduced abundance of *Prevotellaceae*, there may be increased permeability of the intestinal barrier and exposure to toxic substances, which may lead to increased synthesis of alpha-synuclein. Again, preclinical evidence and cross-sectional studies in humans point to abnormal gut microbiota as a key factor in the onset and progression of Parkinson's disease, describing the presence of a specific gut microbiota profile in association with disease and symptom modulation [85,86].

Changes in the gut microbiota have also been correlated with brain disease and peripheral inflammation in patients with Alzheimer's disease and other cognitive disorders [87]. The above is mainly relevant in women, as epidemiological data show they are the predominant risk group for cognitive disorders [77,78]. In their review, Grajek M. et al. collected the results of the most important studies on the psychoprotective effect of probiotics [88].

Probiotics, prebiotics, postbiotics, and the implementation of a balanced diet have been shown to be beneficial in correcting dysbiosis that contributes to obesity [89]. Beneficial actions of probiotics and prebiotics include protection against colonization, stimulation of the production of beneficial bacteria and SCFAs, influence on intestinal transit time metabolism of bile acid salts, participation in vitamin production, modulation the of immune response, and production of specific bioactive substances. The properties of probiotics are strictly strain-dependent, and each strain requires separate studies to determine its properties and efficacy in a specific clinical situation [90]. Numerous studies in both animal and human models confirm the effectiveness of supplementation with probiotics from the genera *Lactobacillus* and *Bifidobacterium*. They favorably affect weight loss processes, seal the intestinal barrier, reduce the intensity of inflammation in the intestine, and have a beneficial effect on reducing visceral fat and adipocyte size. In an animal model study of rats fed a high-fat diet and supplemented with the *Bifidobacterium longum* strain, improvements in immune system performance and glucose tolerance were noted. Another study in an animal model shows the beneficial effects of supplementation with *Lactobacillus* strains (*L. rhamnosus*, *L. plantarum*, *L. gasseri*, *L. fermentum*, *L. reuteri*, *L. paracasei*, and *L. acidophilus*). At different times of use (from 8 days to 12 weeks), a reduction in body weight and visceral fat, and improvement in parameters of lipid and carbohydrate metabolism were demonstrated. An important problem in the treatment of obesity is the common insulin resistance diagnosed in the female population. *Lactobacillus* microorganisms have been found to improve insulin sensitivity by affecting the expression of leptin and fatty acid synthetase, stimulating fatty acid oxidation, and inhibiting lipoprotein lipase activity [91]. In a study by Kadooka et al., supplementation of *Lactobacillus gasseri* SBT2055 strains was used in obese patients. After 12 weeks, a reduction in body weight, visceral and subcutaneous fat, BMI, and waist and hip circumference and an increase in serum adiponectin levels were observed [92,93].

Postbiotics are metabolic byproducts of probiotic microorganisms that exhibit biological activity in the host. Cavallari et al. found that a muramyl dipeptide derived from the bacterial cell wall acts as an insulin-sensitizing postbiotic and can reduce insulin resistance in an animal model of obesity in mice [94]. A study by Dewulf et al. used prebiotic supplementation with inulin-type fructans (ITFs) in women with obesity. Treatment with ITF prebiotics led to an increase in *Bifidobacterium* and *Faecalibacterium prausnitzii* compared to the control group. Both bacteria negatively correlated with serum lipopolysaccharide levels. A decrease in glycemic values after an oral glucose load test was observed in

the group of women taking the probiotic compared to the control group. The use of ITF prebiotics also reduced the abundance of *Bacteroides intestinalis*, *Bacteroides vulgatus* and *Propionibacterium*, which was associated with a slight decrease in fat mass. It has been shown that implementation of ITF prebiotics can help delay or prevent obesity-related comorbidities [95].

One method of altering the composition of the intestinal microbiota and treating obesity is gut microbiota transplantation. Many animal model studies have confirmed the benefits of gut microbiota transplantation in the treatment of obesity. It should be noted that studies of this type have not been conducted in sufficient numbers on humans to conclusively establish the benefits of this treatment method [91]. A study by Vrieze et al. showed that patients with metabolic syndrome after gut microbiota transplantation from healthy donors showed increased insulin sensitivity at 6 weeks after transplantation but without accompanying changes in body weight [96].

A study by Palleja et. al. [97] compared patients before and after bariatric surgery. These authors examined not only weight loss and improvements in the glycemic profile but also changes in the gut microbiota, including changes in the diversity and composition of the microbiota in the three months after surgery. In addition, more than half of the altered microbiota species persisted over the longer term, indicating that bariatric surgery can lead to rapid and permanent changes in patients' gut microbiota. Analysis of the composition of the human fecal microbiome showed that there are six major phyla: *Bacteroidetes*, *Firmicutes*, *Proteobacteria*, *Actinobacteria*, *Fusobacteria*, and *Verrucomicrobia* [98]. Some studies have compared the microbiota of patients before and after surgery (RYGB) to the microbiota of control subjects without surgery to detect changes in gut bacteria after surgery; more specifically, an increase in *Proteobacteria* and *Bacteroidetes* and a decrease in *Firmicutes*. This research aimed to better understand the interaction between the microbiota and obesity and possible ways to modulate the gut microbiota that may benefit patients after bariatric surgery in the future [99].

Regeneration of the microbiota can begin with a change in dietary style. Restoring the pattern represents a stage that cannot be achieved without a difference in mental level through dietary restrictions and limitations. The microbiota pattern can only be adjusted to a small degree by changing the metabolic response, representing the response to the plasticity of the microbiome. Weight loss is a gradual effect that correlates with the presence of biomarkers that modulate the physiological response. The balance of the gut–brain axis is important in establishing homeostasis through neurotransmitters. The metabolic response is mediated by the presence of bioactive compounds in the diet that regulate the synthesis of critical metabolites, such as SCFAs. They are stimulated by a high intake of polyphenols, represent a new direction after future in vitro/in vivo studies, and precisely define clinical relevance [89].

*3.8. Use of Probiotic Therapy in Improving Women's Health*

Based on the fact that the state of the microbiome is very crucial for women's health, it is worth considering the possibility of improving it. In recent years, the issue of targeted probiotic therapy has been a popular and important research topic. Table 1 summarizes selected studies on the impact of probiotic therapy on women's health.

**Table 1.** Review of selected studies on the impact of probiotics and other interventions to regulate the microbiota on women's health.

| Source | Sample | Probiotic Ingredient or Other Intervention | Effect of Therapy |
|---|---|---|---|
| Takahashi et al. [100] | An open-label pilot trial evaluating the safety of probiotic supplementation in lactating women with a 2-month history of allergies. | *L. casei*, *B. longum*, *B. coagulans* | Probiotic supplementation may affect TGF-β levels in human milk while finding a positive effect of probiotic supplementation requires further research. |
| Qiu et al. [101] | Systematic review and meta-analysis were conducted to evaluate the efficacy and safety of probiotics in the prevention of radiotherapy-induced diarrhea in patients with cervical cancer. | Mainly bacteria of the *Lactobacillus* and *Bifidobacterium* species | Probiotic supplementation may reduce the incidence of radiotherapy-induced diarrhea in cervical cancer patients. |
| Shafie et al. [102] | A triple-blind randomized controlled trial was conducted on 66 postmenopausal women aged 45–55 years. | *B. lactis*, *L. acidophilus* | There were improvements in anxiety, stress, and quality of life in postmenopausal women. |
| Husain et al. [103] | Randomized, double-blind, placebo-controlled trial conducted among women aged 16 years or older recruited at 9–14 weeks gestation. | *L. rhamnosus* GR-1, *L. reuteri* RC-14 | Probiotics taken orally from early pregnancy did not modify the vaginal microbiota. |
| van de Wijgert et al. [104] | A systematic review evaluating the effect of vaginal probiotics on the cure and/or recurrence of bacterial vaginosis and vulvovaginal candidiasis. | *Lactobacillus* strains | Probiotics are promising for the treatment and prevention of bacterial vaginosis, but much less so for the treatment and prevention of vulvovaginal candidiasis. |
| Zheng et al. [105] | Review article evaluating the effects of probiotics supplementation on metabolic health and pregnancy complications in pregnant women. | Mainly bacteria of the *Lactobacillus* and *Bifidobacterium* species | Probiotic supplementation during pregnancy has beneficial effects on glucose metabolism but not lipid metabolism among pregnant women. |
| Martoni et al. [106] | A pilot clinical study investigating the clinical effects of a 10-strain probiotic on parameters of vaginal health in women with intermediate Nugent score or vaginal pH > 4.5. | *L. acidophilus* DDS-1, *L. gasseri* UALg-05, *L. plantarum* UALp-05, *L. rhamnosus* UALr-06, *L. reuteri* UALre-16, *L. paracasei* UALpc-04, *L. crispatus* UALcr-35, *L. brevis* UALbr-02, *B. longum* subsp. *longum* UABl-14, *B. animalis* subsp. *lactis* UABla-12 | The probiotic product tested helped to significantly lower vaginal pH in women with intermediate Nugent score or elevated vaginal pH. |

**Table 1.** *Cont.*

| Source | Sample | Probiotic Ingredient or Other Intervention | Effect of Therapy |
|---|---|---|---|
| Sarkar et al. [76] | Review on the role of microbiota and probiotics in neurodegenerative diseases. | *Lactobacillus casei shirota*, *Bacillus* spp. | Regular consumption of a probiotic beverage containing *Lactobacillus casei shirota* has a positive effect on the gut microbiota in patients with Parkinson's disease, while *Bacillus* spp. may have a positive effect on dopamine synthesis. |
| Cenit et al. [78] | Review the role of the gut microbiota in brain development and function. | *Lactobacillus rhamnosus*, *Lactobacillus helveticus*, *Bifidobacterium infantis*, *Bifidobacterium longum*, *Bifidobacterium breve* | Probiotic therapies using the aforementioned strains had an effect on relieving depressive symptoms. |
| Luan et al. [80] | Review of recent metabolomic research findings on the metabolic pathways that exist between the gut microbiota and the brain. | *Lactobacillus* and *Bifidobacterium* species | *Lactobacillus* and *Bifidobacterium* can produce gamma-aminobutyric acid (GABA), which positively affects the exchange of signals between neurons. |
| Banerjee et al. [85] | Review article evaluating the role of gut microbiota in pathogenesis of various neurological conditions. | *Bifidobacterium infantis*, *Bifidobacterium* spp., *Bacillus* spp., *Lactobacillus* spp., *Streptococcus*, and *Enterococcus* spp. | *Bifidobacterium infantis* increases plasma tryptophan, which upregulates serotonin; *Bifidobacterium* spp. synthesise GABA, *Bacillus* spp. synthesize norepinephrine and dopamine, *Lactobacillus* spp. synthesize acetylcholine, *Streptococcus*, and *Enterococcus* spp. produce serotonin. Probiotic therapy could therefore affect mood and cognitive function. |
| Steenbergen et al. [107] | A triple-blind, placebo-controlled study of 20 healthy participants without current mood disorders who received a 4-week intervention with multispecies probiotic foods and 20 control participants receiving a placebo. | *Bifidobacterium bifidum* W23, *Bifidobacterium lactis* W52, *Lactobacillus acidophilus* W37, *Lactobacillus brevis* W63, *Lactobacillus casei* W56, *Lactobacillus salivarius* W24 *and Lactococcus lactis* (W19 and W58) | Probiotic therapy improved mood in depressed patients and reduced negative thoughts. |
| Grajek et al. [88] | Review article on the impact of lifestyle and nutrition on mental health. | *Lactobacillus helveticus*, *Bifidobacterium longum* | The additional use of psychobiotics may prove effective in the treatment of anxiety or depressive disorders. |
| Kadooka et al. [88] | Multicenter, double-blind, randomized, placebo-controlled intervention trial on 87 subjects with higher body mass index and abdominal visceral fat area. | *Lactobacillus gasseri* SBT2055 | After 12 weeks, a reduction in body weight, visceral and subcutaneous fat, BMI, waist and hip circumference, and an increase in serum adiponectin levels were observed. |

**Table 1.** *Cont.*

| Source | Sample | Probiotic Ingredient or Other Intervention | Effect of Therapy |
|---|---|---|---|
| Dewulf et al. [95] | A double-blind, placebo-controlled, intervention study that used prebiotic supplementation with inulin-type fructans (ITFs) in women with obesity. | Inulin/oligofructose 50/50 mix (prebiotic) | The use of ITF prebiotics also reduced the abundance of *Bacteroides intestinalis*, *Bacteroides vulgatus*, and *Propionibacterium*, which was associated with a slight decrease in fat mass. It has been shown that the implementation of ITF prebiotics can help delay or prevent obesity-related comorbidities. |
| Vamanu et al. [89] | Review article on the alleviation of human dysbiosis in degenerative diseases and obesity. | *Lactobacillus curvatus* HY7601, *Lactobacillus plantarum* KY1032; *Lactobacillus reuteri* | The therapy has resulted in the regulation of pro-inflammatory genes in adipose tissue and fatty acid oxidation genes in the liver. *Lactobacillus reuteri* has anti-inflammatory effects due to its role in controlling interleukin (IL)-10 cytokine synthesis. |

In women with healthy physiological conditions, the composition of the microbiota varies. There is a predominance of strains that enable the maintenance of health under changing external conditions, including various stress and inflammation events. In contrast, under conditions of dysbiosis, the microbiota is less diverse and the number of commensal bacteria is too low to allow the maintenance of internal homeostasis. Abnormalities in the intestinal ecosystem can lead to the development of diseases such as obesity, diabetes, inflammatory bowel disease, and certain types of cancer [108]. The capabilities of today's diagnostics and medicine make it possible to treat the microbiome and microbiota as a specific biomarker to implement appropriate therapeutic strategies (Table 2). Individual conditions related to a woman's microbiota may even be responsible for the effectiveness of implemented pharmacotherapy in many diseases and disorders. Interestingly, the bacterial peptides of a woman's microbiota can interfere with disease-affected cells, affecting signaling processes and, ultimately, their proliferation. Moreover, the composition of the microbiota can determine the activity of certain complexes that act as transcription factors. The proportions of individual bacterial species can determine the course of diseases. The presence of *Akkermansia*, *Faecalibacterium*, and *Bifidobacterium* bacteria correlates with anti-inflammatory effects and can reduce the exacerbation of inflammatory bowel diseases, for example [109].

**Table 2.** Potential opportunities for personalized diagnostic and therapeutic strategies.

| **Non-invasive Biomarkers to Help Diagnose and Stage of Disease** |
|:---:|
| identification of women at risk |
| determination of the disease phenotype |
| **Treatment** |
| Intestinal barrier integrity (signaling for toll-like receptors, TLRs) |
| Modulation of intestinal dysbiosis |
| Antimicrobial and antifungal agents |
| prebiotics |
| probiotics |
| synbiotics |
| fecal microbiota transplantation (FMT) |
| bacteriophage therapy |
| Effects on the metabolism of the intestinal microbiota |
| postbiotics |
| molecule inhibition |
| genetically modified microbes |
| Personalized diet therapy |
| **Pharmacomicrobiomics** |
| appropriate selection of pharmaceuticals |

## 4. Strengths and Limitations

There are still few papers in the scientific space summarizing the most important findings related to the impact of microbiota on women's health, especially, as this review does, emphasizing the importance of maintaining homeostasis in the prevention of diseases and disorders. The primary limitation of the presented review of research on the relationship between microbiota and women's health is the plethora of studies on the topic. The multitude of studies here does not mean that they all address the issue presented in

this manuscript. Many of the papers that were searched and included in the review assume a relationship between the microbiota and health, not always taking into account a variable such as a gender, and these are usually very superficial opinions that are not scientifically based or are confirmed on an animal model. The authors are aware that in the face of such a large number of studies, important reports may have been overlooked, but it should be noted that every effort was made to ensure that this review was conducted fairly, taking into account large, multi-center research projects and highlighting mainstream research.

## 5. Conclusions

However, it should be noted that in recent years there have been significant advances in research on the relationship between gender and microbiota modulation. Gender differences are due to several variables among others related to sex hormones, body weight, and physiological and pathological conditions in women. We have attempted to summarize recent studies on this topic in various conditions, such as hormonal changes, aging, inflammatory and functional diseases of the gastrointestinal tract, and their changes in different parts of the gastrointestinal tract. However, most of the studies conducted so far refer to animal models. Further studies of the interaction between gender and gut microbiota may suggest new preventive measures for relevant diseases. Modulations of the microbiota using pre- and probiotics, including symbioses in the female population will provide opportunities to improve female health. Bacterial metabolites could serve as biomarkers for the development of specific metabolic disorders and diseases. This would allow the planning of appropriate strategies for preventive as well as therapeutic action. The microbiome and its specifics could also serve as indicators for predicting the course of a disease, its phenotype, and its potential response to implemented treatment. Modern diagnostics address the diverse needs of women of all ages and conditions so that planned strategies can address individual conditions to the greatest extent possible. Whole-genome sequencing makes it possible to assess the percentage of individual microorganisms, and nanopore technology accurately distinguishes between each bacterial species. This approach is more likely to facilitate the personalization of both dietary recommendations and the use of targeted probiotic therapy. Appropriate targeted probiotic therapy could be, among other things, part of the therapy in the treatment of infertility in women as well as of diseases of endocrine origin. Given the above, it is important to continue research on the microbiome and the implementation of probiotic therapy in the treatment of various disease entities as well as to educate the public about leading a lifestyle that promotes a favorable composition of the microbiome. From a future perspective, the correlation between the bioactivity of the microbiota and the bioavailability of functional compounds should be considered, as it is important to modulate women's well-being.

**Author Contributions:** Conceptualization, K.K.-K.; methodology, K.K.-K., M.G., P.H. and W.G.; formal analysis, K.K.-K., M.G., P.H. and W.G.; investigation, K.K.-K., M.G., P.H. and W.G.; resources, K.K.-K., M.G., P.H. and W.G.; writing—original draft preparation, K.K.-K., P.H. and W.G.; writing—review and editing, K.K.-K., M.G., P.H. and W.G.; supervision, M.G.; project administration, K.K.-K.; funding acquisition, K.K.-K. All authors have read and agreed to the published version of the manuscript.

**Funding:** This research received no external funding.

**Institutional Review Board Statement:** The research complies with the provisions of the Helsinki Declaration.

**Informed Consent Statement:** Not applicable.

**Data Availability Statement:** Not applicable.

**Conflicts of Interest:** The authors declare that they have no conflict of interest.

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
