# Peer review of "The Importance of the Microbiota in Shaping Women’s Health—The Current State of Knowledge"

_2673-8007, doi:10.3390/applmicrobiol3010002_

Round 1

Reviewer 1 Report

The manuscript by Karolina Krupa-Kotara et al. summarized the importance of the microbiota in shaping women's health. This study is of interest to the readers and I have the following suggestions and comments:

1, In the introduction part, I think the authors should discuss more about the human microbiome in different body sites. For example, the composition and the function of the microbiota etc. 

2, The authors focused on the women's health. In this regard, I suggest the authors to disccuss more about the effects of hormons on the human microbiome. Previous studies have indicated that the function of the microbiome is affected by the hormons levels. This should be included in this mini-review. 

3, The authors should discuss more about the future developments in this field. For example, what are the limitations of the current studies and what should we do to advance our understandings of the functions of the microbiome in women's health. 

4, The authors should include more clinical trails about the functions of the microbiota in shaping women's health. The authors should check this website https://clinicaltrials.gov/ for more information. Besides, more studies should be discussed and included in this review. Generally, less than 50 references is not accepatable. 

Author Response

Dear Reviewer,
We appreciate your interest in our manuscript and your valuable suggestions to improve it.
In this manuscript, we provide point-by-point comments highlighted in blue. The revision was developed in consultation with all co-authors and each author has agreed to the final form.
Once again, we thank you for your comments and hope that our manuscript now meets all the requirements for acceptance for publication.

Best regards, Authors

Reviewer 2 Report

Dear authors,

After the review process, I have several comments: in section 3.6, you should expand the presentation with new findings related to the link between obesity, microbiota dysbiosis and neurodegenerative pathologies because now it is a high incidence in the case of old women; as a future perspective, you should include correlation between microbiota bioactivity and bioavailability of functional compounds because it is important to modulate woman well-being.

Best regards!

Author Response

(The authors gave the same response as above.)

Round 2

Reviewer 1 Report

The authors have revised the manuscript accordingly. It can be considered for publication. 

Author Response

Dear Reviewer,
Thank you very much for the time you took to read our manuscript and the many valuable comments to improve it. We have still made additional changes (highlighted in orange) based on the comments of the second reviewer. Thank you for recognizing our work and allowing it to proceed through the editorial process. 
With best regards, Authors

Reviewer 2 Report

Dear authors,

The reference section should comply with my first comments.

Best regards!

Author Response

Dear Reviewer,
Thank you very much for the time you took to read our manuscript and the many valuable comments to improve it. Previous comments, may not have been highlighted enough in the manuscript, so we decided to rewrite it and based on your previous comments, we marked any modifications in orange. We hope we have met your expectations, if not, we kindly ask for additional suggestions on how we can improve our manuscript. 
With best regards, Authors